# Chemical Profiling, Insecticidal, and Phytotoxic Effect of Essential Oils from Leaves and Inflorescence of Moroccan *Chenopodium ambrosioides* (L.)

**DOI:** 10.3390/plants13040483

**Published:** 2024-02-08

**Authors:** Ayoub Kasrati, El Hassan Sakar, Ahmed Aljaiyash, Aziz Hirri, Lahcen Tamegart, Imane Abbad, Chaima Alaoui Jamali

**Affiliations:** 1Laboratory of Drugs Sciences, Biomedical Research and Biotechnology, Faculty of Medicine and Pharmacy, Hassan II University of Casablanca, B. P. 9154, Casablanca 20250, Morocco; aziz.hirri@univh2c.ma; 2Laboratory of Biology, Ecology, and Health, FS, Abdelmalek Essaadi University, Tetouan 93002, Morocco; e.sakar@uae.ac.ma; 3Faculty of Pharmacy, Omar Al-Mukhtar University, Albayda 0463, Libya; aaljaiyash@yahoo.com; 4Laboratory of Engineering in Chemistry and Physics of Matter, Faculty of Sciences and Technics, Sultan Moulay Slimane University, Beni Mellal 23040, Morocco; 5Department of Biology, Faculty of Science, Abdelmalek Essaadi University, Tetouan 93000, Morocco; l.tamegart@uae.ac.ma; 6Faculty of Sciences Semlalia, University Cadi Ayyad, Marrakech 40000, Morocco; imane.abbadd@gmail.com; 7Laboratory of Environmental Biology and Sustainable Development, Ecole Normale Supérieure, Abdelmalek Essaadi University, Tetouan 93000, Morocco; chaima.tw@gmail.com

**Keywords:** *Chenopodium ambrosioides* (L.), essential oils, chemical composition, phytotoxic activity, insecticidal activity

## Abstract

*Chenopodium ambrosioides* (L.) is a medicinal and aromatic plant widely used in the Moroccan traditional medicine for its important pharmacological properties. In order to contribute to the valorization of this plant, the current study aimed at evaluating, for the first time, the variation in the yield, chemical composition, and insecticidal and phytotoxic activities of *C. ambrosioides* essential oil (CAEO) isolated from leaves and inflorescences. The results obtained showed that the CAEO yields vary significantly according to the distilled plant part, with 0.75 ± 0.15% for the leaves and 1.2 ± 0.34% for the inflorescences. CAEO profiling showed quantitative more than qualitative differences. Leaf CAEO was rich in δ-3-carene (61.51%), followed by p-cymene (14.67%) and 1,2:3,4-diepoxy-p-menthane (6.19%). However, inflorescence CAEO was dominated by the same compounds but with variable levels (δ-3-Carene: 44.29%; 1,2:3,4-diepoxy-p-menthane: 19.46%; and p-cymene: 17.85%). The CAEOs from the leaves and inflorescences showed a very interesting inhibiting effect on the germination and growth of the three species used, namely *Medicago sativa*, *Linum rusitatissimum*, and *Raphanus sativus*. However, there was no significant difference between these CAEOs. Similarly, for the insecticidal activity, CAEOs exhibited an important and similar toxicity against *Tribolium confusum* adults with LD_50_ of 4.30 and 4.46 μL/L of air and LD_90_ of 6.51 and 9.62 μL/L air for toxicity by fumigation, while values for contact toxicity on filter paper were 0.04 and 0.05 μL/cm^2^ for LD_50_ and 0.08 and 0.09 μL/cm^2^ for LD_90_.

## 1. Introduction

Within the Chenopodiaceae family, the genus *Chenopodium* has been considered one of the largest genera in this family, with an estimated number of around 1400 species [1]. This genus includes spontaneous and cultivated species widely distributed across Europe, Africa, Asia, Australia, and North America. Species in this genus are herbaceous and perennial plants found in coastal saline environments, plains, high-altitude valleys (Andes, Himalayas), and barren deserts (Atacama, Australia) [2]. Since ancient times, many species of *Chenopodium* genus have been used in a wide variety of ways, and play a significant role in therapeutic applications. Indeed, the aerial parts of these species are commonly used in form of powders and decoctions to treat cold illnesses, abdominal pain, and leishmaniasis, and also as anti-inflammatories, antiasthmatics, antipyretics, antispasmodics, hypotensives, and carminatives [3,4,5]. Concerning the chemical composition, most species of the *Chenopodium* genus owe their activities and properties to their contents of several secondary metabolites, especially essential oils rich in terpenoids. Several studies have shown that these substances have significant antimicrobial, antioxidant, and insecticidal activities [6,7,8].

In Morocco, the genus *Chenopodium* is represented by 10 spontaneous and introduced species [9]. Among these species, *Chenopodium ambrosioides*, commonly known as Mexican tea and “M’khinza” in Arabic, is an annual plant with a strong aromatic odor, reaching up 0.3 to 1 m in height. This species grows in uncultivated fields, along roadsides and in abandoned places [9]. *C. ambrosioides* is among the species most frequently used in Moroccan folk medicine. In fact, the species is used as an infusion for the treatment of various illnesses, including gastrointestinal disorders, typhoid, dysentery in children and adults, and fever [10]. Many pharmacological studies supported the cytostatic, hypotensive, anti-inflammatory, analgesic, and antipyretic activities of *C. ambrosioides* crude extracts [10,11]. On the other hand, it has been reported that the essential oils isolated from the aerial parts of *C. ambrosioides* exhibit a variety of biological activities including antimicrobial, antioxidant, and acaricidal activities [12,13,14,15]. These properties have been ascribed to the presence of numerous monoterpenes including, α-terpinene, ascaridole, carvacrol, p-cymene, and o-cymene [12,13,14,15].

As known, the yield and chemical composition of essential oils isolated from several medicinal and aromatic plants, and consequently their biological activities, are influenced by several factors, including the part of the plant used for distillation [16,17,18]. However, to our knowledge, no work has been published on the effect of this factor on the quantity and quality of *C. ambrosioides* essential oils (CAEO). In this context, the aim of the present study is to evaluate for the first time the variation in yield, chemical composition, insecticidal effects, and phytotoxic activities of CAEO isolated from the leaves and inflorescence. The insecticidal properties were evaluated against the common pest *Tribolium confusum* known for attacking and infesting stored flour and grain. The phytotoxic effects were determined in seeds obtained from three species, namely *Medicago sativa*, *Linum usitatissimum*, and *Raphanus sativus*, widely used to determine the phytotoxic activity of essential oils.

## 2. Results and Discussion

### 2.1. CAEO Yield

Essential oils obtained from *C. ambrosioides* leaves and inflorescences were characterized by an orange color with an unpleasant odor. Average essential oil yields (*v*/*w* on dry weight basis) are shown in Table 1. The results indicated that the essential oil yields varied according to the plant part, with 0.75 ± 0.15% for leaves and 1.2 ± 0.34% for inflorescences. The essential oil yield obtained in this study for leaves is comparable to that found by other authors [19,20]. To our knowledge, no data have been published on the essential oil yield of *C. ambrosioides* inflorescences. According to the literature, the essential oil yield of several aromatic and medicinal plants varies depending on plant material parts. Indeed, according to [13], the yield of essential oil from *Coriandrum sativum* leaves (0.1%) is lower than that recorded for seeds of the same species (1.1%). Another study carried out on *Hippophaer hamnoides* showed that essential oil yields differed between seeds, leaves, and pulp (0.033, 0.365, and 1.26%, respectively) [14]. This phenomenon can be explained by differences in the enzymatic activity of the different parts of the plant, resulting in a notable variation in the synthesis of volatile compounds [21].

### 2.2. Chemical Composition of CAEOs

The essential oils obtained from the different samples were subjected to detailed GC/MS analysis in order to determine different volatiles in our CAEOs according to plant part. The general chemical profiles of the tested oils (percentage content of the individual components, retention time, retention index, and chemical class distribution) are summarized in Table 1. A representative chromatogram for CAEOs isolated from leaves and inflorescences is depicted in Figure 1.

In total, 10 compounds were identified, representing 97.54 and 95% of total constituents, respectively. Essential oils extracted from *C. ambrosioides* leaves and inflorescences were characterized by a high concentration of hydrocarbon monoterpenes (78.01–63.26%), the main compounds being δ-3-carene (61.51–44.29%), p-cymene (14.67–17.85%) and 1,2:3,4-diepoxy-p-menthane (6.19–19.46%). These results differ from those obtained for the same species collected in Morocco in the Meknes and Rhamna region [11,22]. In these studies, CAEOs were characterized mainly by a high content of α-terpinene (61.04% and 23.77%, respectively). Our CAEOs differ from those described in the literature by a high content of δ-3-carene and an absence of α-terpinene. These results suggest that our studied essential oil can be considered a δ-3-carene chemotype.

A comparison of the chemical composition of essential oils extracted from leaves and inflorescences of *C. ambrosioides* showed quantitative rather than qualitative differences. Indeed, the oil obtained from leaves showed high levels of δ-3-carene (61.51%), followed by p-cymene (14.67%) and 1,2:3,4-diepoxy-p-menthane (8.15%), while that obtained from inflorescences was characterized by the dominance of δ-3-carene (44.29%), followed by 1,4-epoxy-p-menth-2-ene (19.46%), and p-cymene (17.85%). This chemical variation between the different parts of *C. ambrosioides* is in line with what has been previously reported for other aromatic and medicinal plants such as *Hippophae rhamnoides*, *Coriandrum sativum*, and *Eucalyptus carnaldulensis* [13,14,23]. These differences in the constituents of essential oil between leaves and inflorescences of *C. ambrosioides* can be related to the influence of several intrinsic factors on the plant biosynthetic pathways of some terpenoids and consequently the relative changes in the number and the content of the main characteristic compounds [23].

### 2.3. Phytotoxic Activity of CAEOs

The phytotoxic effect of *CAEOs* on *Medicago sativa*, *Linum rusitatissimum*, and *Raphanus sativus* seeds was assessed by determining the germination parameters (germination percentage (GP) and mean time of germination (MGT)) and growth parameters (size, fresh weight (FW), dry weight (DW), and vigor index (VI)), as presented in Table 2, Table 3 and Table 4. The results obtained showed that these oils have a strong and variable inhibitory effect on germination and growth of all the seeds studied. This variation depends essentially on the seed and the concentration used.

The germination results showed that there was no significant difference (*p* < 0.05) between the effect of the essential oils studied on the seeds tested. These data also showed that *L. rusitatissimum* and *A. sativus* seeds were the most sensitive compared with *M. sativa.* All CAEOs concentrations showed a lower germination percentage (GP) and a longer mean germination time (MGT) than the control. The high concentrations of 1 and 0.5 µL/mL expressed a total inhibitory effect on *L. rusitatissimum* and *A. sativus* seeds. However, this total inhibitory effect on germination was only observed at a concentration of 1 µL/mL for *M. sativa* seeds.

With regard to growth, both CAEOs showed a significant negative effect with a non-significant difference (*p* < 0.05) on seedlings of the species studied. This effect increased with the concentration used. The results showed that *M. sativa* and *A. sativus* seedlings were the most resistant to the effect of CAEOs, since the decrease in growth parameters compared to control was only observed at the 0.25 µL/mL concentration. However, this growth-inhibiting action was recorded at the lowest concentration (0.125 µL/mL) for *L. rusitatissimum* seedlings.

The phytotoxic potency of CAEOs observed in this study is comparable to what has been reported in the literature on this species [24,25] and justifies the considerable potential of this plant as a bio-herbicide. The phytotoxic activity of these oils can be attributed to the presence of high concentrations of two hydrocarbon monoterpenes, δ-3-carene and p-cymene. Indeed, several studies have reported that hydrocarbon monoterpenes present in essential oils are known for their important phytotoxic effect [26,27]. This activity could be explained by the ability of these compounds to disrupt germination as well as radicle growth, to reduce proteins and nucleic acids, and to cause alterations in ion absorption, water balance, phytohormonal balance, photosynthesis, and respiration [16,28,29].

### 2.4. Insecticidal Activity of CAEOs

The insecticidal activity of CAEOs was assessed by two tests, namely contact toxicity on filter paper and fumigant toxicity. The percentage of mortality (%) and lethal dose values (LD_50_ and LD_90_) are summarized in Table 5 and Table 6. The data showed that EOs tested exhibited an interesting toxicity towards adults of *T. confusum.* Moreover, in the treatment of essential oils against this insect, mortality was a concentration-dependent response (Table 5). In fact, the mortality percentages increased with increasing the concentration of the oils, reaching the highest values in the highest applied doses. The mortality percentage values ranged from 13.33 to 82.66% and from 12.33 to 100%, for contact and fumigant toxicity assays, respectively. Based on Table 6, the results obtained from both tests showed that there was no significant difference between leaf and inflorescence oils against *T. confusum,* since no overlap between the confidence limits was observed. For the fumigant assay, the values obtained were in order of 4.30–4.46 µL/L of air for LD_50_ and 6.51–9.62 µL /L of air for LD_90._ For the contact assay, the LD_50_ values obtained are 0.04 and 0.05 UL/cm^2^, and for the LD_90_ values are 0.08 and 0.09 µL/cm^2^. These results are in line with other work on the insecticidal activity of essential oils of the same species against several insect pests of stored food products [19,30,31].

Generally, the insecticidal activity of essential oils isolated from both leaves and inflorescences can be ascribed to their chemical composition and especially to their major compounds and synergetic effects cannot be discarded. Thus, the important toxic effect of CAEOs against *T. confusum* adults is probably due to their high content of δ-3-carene, a hydrocarbon monoterpene well documented for its acetylcholinesterase inhibitory activity, hence its strong insecticidal activity [32]. The toxicity of these oils can also be attributed to the presence of significant quantities of p-cymene. The insecticidal effect of this compound has previously been demonstrated against various insect pests of stored food products [33,34]. In addition, the components in lower amounts may also contribute to insecticidal activity of the essential oils, likely involving some type of synergism with other active compounds [33].

## 3. Materials and Methods

### 3.1. Plant Material

The aerial parts of *C. ambrosioides* were collected at the flowering stage during 2020 from the Marrakech region (Wahate sidi Ibrahim, 31°30′ N/07°40′ W). This species was identified by Prof. Abdelaziz Abbad at the Biology Department, Faculty of Sciences Semlalia, Marrakech. Freshly harvested plant parts (leaves and inflorescences) were cut separately into small pieces to facilitate extraction of the essential oils. Both parts were then dried at room temperature, protected from light and humidity.

### 3.2. Essential Oils Isolation

Essential oils were obtained by hydro-distillation using a Clevenger-type apparatus. Each dried part (leaves and inflorescences, 200 g) was distilled for 4 h. At the end of extraction, we noticed that an important amount of essential oil remained in the hydrolate. In order to recover this amount, the mixture hydrolate/essential oil was subjected to a second solvent extraction using dichloromethane. The dichloromethane was chosen thanks to its low boiling point (40 °C). The lower phase (essential oil + dichloromethane) recovered was then subjected to evaporation (removal of the extraction solvent) using a rotary evaporator at a low temperature to remove completely the solvent used but also to avoid any changes in our oil chemical profile. The recovered essential oil was dried with anhydrous sodium sulfate, then stored at 4 °C in the dark.

### 3.3. Chromatographic Analysis of Essential Oils

The analytical GC/MS system used was an Agilent 6890/5973 GC/MSD system (Agilent Technologies, Palo Alto, CA, USA) equipped with an Agilent DB-5ms cap. column (30.0 m 0.25 mm i.d., film thickness 0.25 mm; model number 122-5532). The oven temp. was programmed to rise from 60 to 246 °C at 3 °C/min; injector, transfer, source, and quadrupole temperatures were 260, 280, 230, and 150 °C, respectively; carrier gas, He (high purity; constant linear velocity of 37 cm/s); injection volume, 1.0 µL (of samples of 60 µL of EO diluted with 2 mL of acetone); split ratio, 1:50; ionization voltage, 70 eV; m/z range, 41–450 amu.

The identification of the individual components was based on (i) the comparison of their mass spectra with those of authentic reference compounds where possible and with those listed in the WILEY275 and NBS75K libraries and Adams terpene library [35] and (ii) the comparison of their retention indices (RIs) determined on a DB5 cap. column (nonpolar, 5% phenyl polysilphenylene-siloxane) relative to the retention times (tR) of a series of n-alkanes (C9–C24) with linear interpolation, with those of authentic compounds or the literature data. For semi-quantification purposes, the normalized peak area of each compound was used without any correction factors, to establish relative contents.

### 3.4. Phytotoxic Activity of Essential Oils

The test was carried out using the method described in [36]. Preliminary testing is essential to define the concentration range to be used for seed treatment. The essential oils were emulsified with Tween 80, at a ratio of 1:1 (*v*/*v*). The mixture was dissolved in distilled water to obtain concentrations of 1, 0.5, 0.25, and 0.125 µL/mL, while a solution of Tween 80 in water was used as a control. Subsequently, aliquot of 5 mL of each concentration was added to glass Petri dish (9 cm) with two layers of filter papers (Whatman No. 1). Seeds of the species tested *Medicago sativa*, *Linum usitatissimum*, and *Raphanus sativus* were sterilized with a sodium hypochlorite solution (1%) for 20 min. The seeds were then rinsed three times with distilled water. Four replicates were prepared for each concentration of the oils, each comprising 20 seeds for each species tested. Petri dishes were sealed with a Parafilm^®^ tape and kept at 27 °C in a dark growth chamber. Germinated seeds (2 mm root length) were counted daily. Germination percentage (GP), mean germination time (MGT), growth parameters (height, fresh weight (FW), dry weight (DW)) and vigor index (VI) were calculated on the seventh day according to the following formulas:Germination percentage (GP) = n/N × 100
Mean germination time (MGT) = ((n.d)/N)
Vigor index (IV) = Seedling length (cm) × PG
where n = number of germinated seeds, N = total number of seeds, and d = number of days.

### 3.5. Insecticidal Activity of Essential Oil

#### 3.5.1. Insect Rearing

Colonies of the brown flour beetle, *T. confusum*. (Coleoptera: Tenebrionidae), were maintained in the laboratory without exposure to insecticides. Sixty insects of both sexes were reared on a mixture of wheat flour, wheat germ, and yeast extract (13:6:1 by *w*/*w*/*w*) in glass containers (16 cm diameter × 22 cm height). All containers were covered with a fine mesh cloth for ventilation. The culture was carried out in a growth chamber at 26 ± 1 °C, with a relative humidity (RH) of 70–85% and 16:8 h light/dark photoperiod. Only young adults were used in the tests. All tests were carried out under conditions identical to those of the cultures. In all bioassays, insects were considered dead when no leg or antennal movements were observed. The bioassays were designed to assess median lethal doses (LD_50_ and LD_90_ values) (doses that killed 50% and 90% of the exposed insects, respectively).

#### 3.5.2. Contact Toxicity on Filter Paper

The test was carried out according to the method described by [37]. Several preliminary tests were carried out to select the doses to be used for CAEOs. Four doses were prepared by diluting 1, 2, 3, and 4 µL of essential oils in 1 mL acetone, corresponding to doses of 0.016, 0.031, 0.050, and 0.063 µL/cm^2^. One mL of each solution was dispensed on a 9 cm diameter (63.62 cm^2^ surface area) filter paper disk (Whatman n°1) placed in a glass Petri dish of the same diameter. For the control, filter papers were treated with acetone only. After 10 min, once the solvent had been evaporated, 10 unsexed adults freshly collected from their rearing environment, aged 7 to 14 days, were introduced into each Petri dish, which were then resealed. Three replicates were performed for e each dose. Mortality was recorded after 24 h. Insects were considered dead if no movement of legs or antennae was recorded.

#### 3.5.3. Fumigant Toxicity of Essential Oils

The test was carried out according to the method described by [37]. Several preliminary tests were carried out to select the doses to be used for each species. Four doses were prepared by depositing respective volumes of 0.125, 0.25, 0.5, and 1 µL of CAEOs on 2 cm-diameter filter papers (Whatman No. 1) attached to the screw caps of 60 mL Plexiglas bottles. These volumes correspond to fumigant concentrations of 2.083, 4.166, 8.333, and 16.666 µL/L of air. For the control, the filter paper square was not impregnated with essential oil. A total of 10 unsexed adults, freshly collected from their rearing environment and aged between 7 and 14 days, were introduced into each bottle, which was then immediately resealed. Three replicates were carried out for each dose. Mortality was recorded after 24 h. Insects were considered dead when no movement of legs or antennae was recorded.

### 3.6. Data Analysis

Measurements and determinations were performed in triplicate and then averaged [38]. The results of phytotoxic activity of the studied essential oils were analyzed using IBM SPSS Statistics version 25 (IBM Corp., Armonk, NY, USA) for analysis of variance (ANOVA). The statistically significant differences were separated using the Student–Newman–Keuls test (*p* < 0.05). Mortality results for different concentrations of CAEOs were transcribed and analyzed by the same software using the probit-log model [39]. The analysis was used to determine LD_50_ and LD_90_ values with their confidence limits and Chi-square (χ^2^).

## 4. Conclusions

In conclusion, the essential oils from *C. ambroisoides* leaves and inflorescences studied showed a quantitative variation in the chemical composition and interesting phytotoxic and insecticidal activities. The results showed that the highest yield of essential oil was observed in the inflorescences. An analysis of the chemical composition of these oils showed quantitative rather than qualitative differences, with the dominance of δ-3-carene, p-cymene, and 1,2:3,4-diepoxy-p-menthane. These oils also showed a significant inhibitory effect, with a non-significant difference in the germination and growth of *Medicago sativa*, *Linum rusitatissimum*, and *Raphanus sativus*. They also showed an interesting and similar toxicity against the *T. confusum* pest. Essential oils from the leaves and inflorescences of *C. ambroisoides* can be recommended as bio-herbicides and bio-insecticides against weeds and food pests, as alternatives to harmful synthetic chemical products. This approach can help to reduce the applied quantity, and subsequently reduce the negative impact of synthetic agents on human health and the environment. For the application of such essential oils for foods conservation, it is imperative to conduct further investigations to determine the possible toxic effects of the studied oils on human health.

## Figures and Tables

**Figure 1 plants-13-00483-f001:**
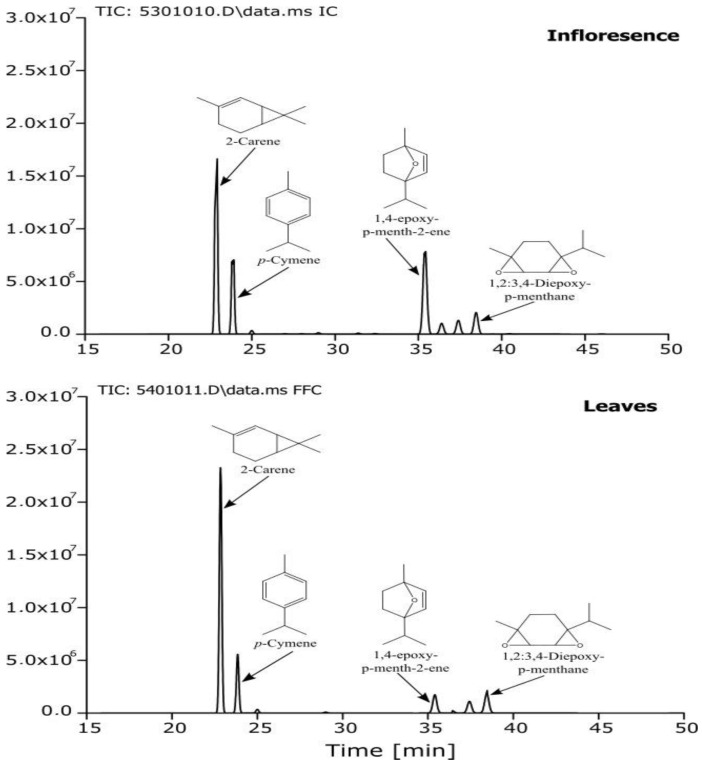
Chromatograms of the identified compounds of the studied CAEOs. Structures of the four major identified compounds are drawn.

**Table 1 plants-13-00483-t001:** Percentage composition of essential oils obtained from leaves and inflorescences of *C. ambrosioides* (L.).

RT	RI	Compound ^a^	Leaves	Inflorescences
22.95	1035	δ-3-Carene	61.51	44.29
23.48	1044	p-Cymene	14.67	17.85
23.63	1048	Limonene	0.64	0.35
23.92	1053	β-Phellandrene	0.12	0.10
25.11	1076	γ-Terpinene	1.07	0.67
35.44	1281	1,4-epoxy-p-menth-2-ene	6.19	19.46
36.06	1294	Ascaridole	0.98	2.36
37.10	1316	Thymol	2.75	2.08
37.62	1327	Carvacrol	1.46	1.54
38.55	1348	1,2:3,4-diepoxy-p-menthane	8.15	6.30
		**Monoterpene hydrocarbons**	78.01	63.26
		**Oxygenated monoterpenes**	19.53	31.74
		**Total (%)**	97.54	95.00
		**Yield (% (*v*/*w*))**	0.75 ± 0.15	1.2 ± 0.34

^a^ Compounds listed in order of elution; RT, retention time; RI, retention indices measured relative to *n*-alkanes (C-9 to C-24) on the non-polar DB-5 column.

**Table 2 plants-13-00483-t002:** Phytotoxic effect of essential oils from the leaves and inflorescences of *C. ambrosioides* on *Medicago sativa* seeds.

Parameters	GP (%)	MTG (Days)	Size (cm)	VI	FW (mg/Plant)	DW (mg/Plant)
**Control**	96.3 ± 4.8 a	4.0 ± 0.0 c	2.6 ± 0.6 a	248.1 ± 6 a	18.5 ± 3.2 a	1.3 ± 0.2 a
**Leaves**						
1 µL/mL	0.0 ± 0.0 c	7.0 ± 0.0 a	0.0 ± 0.0 d	0.0 ± 0.0 c	0.0 ± 0.0 c	0.0 ± 0.0 c
0.5 µL/mL	62.5 ± 6.5 b	4.4 ± 0.1 b	0.6 ± 0.1 cd	39.3 ± 8.9 c	8.5 ± 1.4 b	0.8 ± 0.1 b
0.25 µL/mL	92.5 ± 2.5 a	4.1 ± 0.1 c	1.1 ± 0.5 bc	100.1 ± 41.3 b	13.6 ± 2.9 a	1.1 ± 0.2 ab
0.125 µL/mL	92.5 ± 5.0 a	4.1 ± 0.0 c	2.6 ± 0.6 a	235.6 ± 43.9 a	17.9 ± 3.5 a	1.2 ± 0.1 ab
**Inflorescence**						
1 µL/mL	0.0 ± 0.0 c	7.0 ± 0.0 a	0.0 ± 0.0 d	0.0 ± 0.0 c	0.0 ± 0.0 c	0.0 ± 0.0 c
0.5 µL/mL	56.3 ± 14.4 b	4.4 ± 0.2 b	0.4 ± 0.0 cd	23.8 ± 4.9 c	7.3 ± 1.1 b	1.1 ± 0.1 ab
0.25 µL/mL	91.3 ± 8.5 a	4.1 ± 0.1 c	1.6 ± 0.2 b	144.4 ± 6.4 b	17.0 ± 2.8 a	1.1 ± 0.1 ab
0.125 µL/mL	93.8 ± 2.5 a	4.1 ± 0.0 c	2.4 ± 0.4 a	218.6 ± 37.8 a	17.4 ± 1.6 a	1.3 ± 0.2 a

**GP**: germination percentage; **MTG**: mean time of germination in days; **VI**: vigor index; **FW**: fresh weight; **DW**: dry weight. Within each column, values followed by the same letter are not significantly different at *p* < 0.05.

**Table 3 plants-13-00483-t003:** Phytotoxic effect of essential oils from the leaves and inflorescences of *C. ambrosioides* on *Linum usitatissimum* seeds.

Parameters	GP (%)	MTG (Days)	Size (cm)	VI	FW (mg/Plant)	DW (mg/Plant)
**Control**	90.0 ± 0.0 a	4.5 ± 0.0 c	13.4 ± 0.6 a	1202.7 ± 53.9 a	54.4 ± 3.5 a	4.8 ± 0.5 a
**Leaves**						
1 µL/mL	0.0 ± 0.0 d	7.0 ± 0.0 a	0.0 ± 0.0 d	0.0 ± 0.0 d	0.0 ± 0.0 d	0.0 ± 0.0 c
0.5 µL/mL	0.0 ± 0.0 d	7.0 ± 0.0 a	0.0 ± 0.0 d	0.0 ± 0.0 d	0.0 ± 0.0 d	0.0 ± 0.0 c
0.25 µL/mL	38.7 ± 8.5 c	4.6 ± 0.1 bc	0.32 ± 0.11 d	12.2 ± 3.8 d	17.7 ± 6.3 c	4.3 ± 0.5 ab
0.125 µL/mL	77.5 ± 2.0 b	4.6 ± 0.0 bc	3.24 ± 0.51 c	249.5 ± 32.6 c	31.7 ± 2.9 b	4.2 ± 0.4 ab
**Inflorescence**						
1 µL/mL	0.0 ± 0.0 d	7.0 ± 0.0 a	0.0 ± 0.0 d	0.0 ± 0.0 d	0.0 ± 0.0 d	0.0 ± 0.0 c
0.5 µL/mL	0.0 ± 0.0 d	7.0 ± 0.0 a	0.0 ± 0.0 d	0.0 ± 0.0 d	0.0 ± 0.0 d	0.0 ± 0.0 c
0.25 µL/mL	43.7 ± 11.1 c	4.8 ± 0.2 b	0.22 ± 0.02 d	9.8 ± 3.1 d	18.7 ± 1.6 c	3.9 ± 0.5 b
0.125 µL/mL	68.7 ± 7.5 b	4.6 ± 0.1 bc	4.46 ± 0.13 b	306.8 ± 38.8 b	33.6 ± 2.7 b	4.7 ± 0.2 a

**GP**: germination percentage; **MTG**: mean time of germination in days; **VI**: vigor index; **FW**: fresh weight; **DW**: dry weight. Within each column, values followed by the same letter are not significantly different at *p* < 0.05.

**Table 4 plants-13-00483-t004:** Phytotoxic effect of essential oils from the leaves and inflorescences of *C. ambrosioides* on *Raphanus sativus* seeds.

Parameters	GP (%)	MTG (Days)	Size (cm)	VI	FW (mg/Plant)	DW (mg/Plant)
**Control**	86.7 ± 4.7 a	4.1 ± 0.0 e	6.2 ± 0.7 bc	537.5 ± 36.8 c	70.9 ± 10.6 a	5.0 ± 0.7 ab
**Leaves**						
1 µL/mL	0.0 ± 0.0 d	7.0 ± 0.0 a	0.0 ± 0.0 d	0.0 ± 0.0 e	0.0 ± 0.0 d	0.0 ± 0.0 c
0.5 µL/mL	0.0 ± 0.0 d	7.0 ± 0.0 a	0.0 ± 0.0 d	0.0 ± 0.0 e	0.0 ± 0.0 d	0.0 ± 0.0 c
0.25 µL/mL	57.5 ± 18.5 b	4.7 ± 0.1 b	5.0 ± 1.5 c	282.2 ± 93.3 d	50.4 ± 8.4 b	5.3 ± 0.9 ab
0.125 µL/mL	90.0 ± 4.1 a	4.3 ± 0.1 c	7.6 ± 1.1 ab	678.0 ± 76.9 b	74.8 ± 2.9 a	6.2 ± 0.6 ab
**Inflorescence**						
1 µL/mL	0.0 ± 0.0 d	7.0 ± 0.0 a	0.0 ± 0.0 d	0.0 ± 0.0 e	0.0 ± 0.0 d	0.0 ± 0.0 c
0.5 µL/mL	0.0 ± 0.0 d	7.0 ± 0.0 a	0.0 ± 0.0 d	0.0 ± 0.0 e	0.0 ± 0.0 d	0.0 ± 0.0 c
0.25 µL/mL	30.0 ± 4.1 c	4.6 ± 0.2 bc	0.6 ± 0.3 d	19.7 ± 11.3 e	19.6 ± 5.6 c	4.1 ± 1.1 b
0.125 µL/mL	92.5 ± 2.9 a	4.4 ± 0.1 cd	8.3 ± 0.5 a	772.8 ± 54.6 a	80.4 ± 4.9 a	7.3 ± 2.7 a

**GP**: germination percentage; **MTG**: mean time of germination in days; **VI**: vigor index; **FW**: fresh weight; **DW**: dry weight. Within each column, values followed by the same letter are not significantly different at *p* < 0.05.

**Table 5 plants-13-00483-t005:** The percentage of mortality (%) of leaves and inflorescences of CAEOs against the adults of *T. confusum* in contact and fumigant toxicity bioassays.

Tests		Mean Mortality (%)	
**Contact** **Toxicity**	**Concentrations** **(µL/cm^2^)**	**Leaves**	**Inflorescences**
	0	0 ± 0	0 ± 0
	0.016	13.33 ± 6.66	17.23 ± 5.25
	0.031	33.33 ± 8.81	39.23 ± 4.12
	0.050	50.33 ± 6.12	60.33 ± 6.58
	0.063	82.66 ± 5.35	80.33 ± 6.66
**Fumigant** **Toxicity**	**Concentrations** **(µL/Lair)**		
	0	0 ± 0	0 ± 0
	2.083	12.33 ± 3.33	16.66 ± 6.66
	4.166	30.33 ± 5.77	36.66 ± 4.23
	8.333	59.33 ± 8.81	66.66 ± 10.01
	16.666	100 ± 0	100 ± 0

**Table 6 plants-13-00483-t006:** LD_50_ and LD_90_ values of essential oils from the leaves and inflorescences of *C. ambrosioides* applied by contact and fumigant toxicity bioassays against *T. confusum*.

Tests	Essential Oils	LD_50_ (95% CL) ^a^	LD_90_ (95% CL)	Slope± SE	Chi Square (χ^2^)	df
Contact (µL/cm^2^)	Leaves	0.05 (0.04–0.09)	0.08 (0.06–0.60)	6.73 ± 2.67	0.02	2
	Inflorescences	0.04 (0.03–00.06)	0.09 (0.06–0.49)	3.77 ± 1.25	1.37	2
Fumigant (µL/Lair)	Leaves	4.46 (2.99–6.18)	9.62 (6.81–22)	3.84 ± 1.01	3.45	2
	Inflorescences	4.30 (3.19–5.65)	6.51 (5.11–14.45)	7.10 ± 2.32	0.28	2

LD: lethal dose; ^a^ confidence limits.

## Data Availability

The manuscript includes all the research data used in this study.

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
