# Peer review of "Chemical Profiling, Insecticidal, and Phytotoxic Effect of Essential Oils from Leaves and Inflorescence of Moroccan Chenopodium ambrosioides (L.)"

_plants, 2024, doi:10.3390/plants13040483_

Round 1
Reviewer 1 Report (Previous Reviewer 2)
Comments and Suggestions for Authors
Dear authors of the manuscript entitled “Chemical Profiling, Insecticidal and Phytotoxic Effect of Essential Oils from Leaves and Inflorescence of Moroccan Chenopodium ambrosioides (L.)”, unfortunately I was not able to access your comments in response to the reviewers (myself included), but I only was able to evaluate the file in review mode. I saw that you followed my instructions, with the exception of explaining how you decided to test the two essential oils against Medicago sativa, Linum rusitatissimum and Aphanus sativus and against Tribolium confusum, since the motivation does not emerge from the text: a sentence should be inserted in the introduction to contextualize the work.
Author Response
Response: Thank you very much for carefully reviewing the details of our manuscript. As peer your recommendation, the information required have been added at the end of the introduction section. Please see the revised version.
Please see the attachment

Reviewer 2 Report (New Reviewer)
Comments and Suggestions for Authors
The authors have indicated many change in the manuscript probably following the previous reviewers' instructions.
Few points still need to be clarified or modified. In general, I invite the authors to apply a greater care in the revision of the manuscript.
Specific comments are reported below:
L84: insert “other authors” after “that founded by”
L86: depending “on” and not “to”
L138, L146, L149 . L343 etc. Aphanus sativus is reported, while in L 181 and L278 Raphanus sativus is reported. Please uniform and correct where necessary.
Table 1: Please correct “Yiled”
L144, L 153: why p<0.05 is reported if the results have no significant difference?
L195: erase “were”
L 199: remove the brackets. They are unnecessary.
L233-241: was the essential oil recovered from the hydrolate mixed to the essential oil obtained by hydrodistillation?
L. 238: removal of the extraction solvent.
L. 281 remove the full stop after “Parafilm”
L336: actually the differences were quantitative, and not “notable”, as the chemical compounds detected were the same, as explained few lines below.
L346: the authors recommend the use of C. ambroisoides EOs against weeds and pests, nevertheless no information about the toxicity against humans are reported. Moreover, the authors specified that the EOs were characterized by orange color and unpleasant odor, therefore in my opinion they could be not easily applied in the food industry. I invite the authors to better explain and motivate their conclusion or to add some considerations about it.
Comments on the Quality of English Language
The manuscript has been revised, nevertheless the English language used is very simple. I kindly invite the authors to use a greater care in the revision of the manuscript, particularly as regards English language and punctuation, before the next resubmission.
Author Response
The authors have indicated many changes in the manuscript probably following the previous reviewers' instructions. Few points still need to be clarified or modified. In general, I invite the authors to apply a greater care in the revision of the manuscript.
Thank you very much for carefully reviewing the details of our manuscript.
Specific comments are reported below:
L84: insert “other authors” after “that founded by”
Response : It has been done
L86: depending “on” and not “to”
Response : It has been done
L138, L146, L149 . L343 etc. Aphanus sativus is reported, while in L 181 and L278 Raphanus sativus is reported. Please uniform and correct where necessary.
Response : It has been done
Table 1: Please correct “Yiled”
Response : It has been done
L144, L 153: why p<0.05 is reported if the results have no significant difference?
Response : The statistically significant differences between the phytotoxic activity of the studied oils is essentially based on a statistical analysis using the Student-Newman-Keuls test (p < 0.05). To the best of our knowledge, the communication of statistical results depends on significance level and therefore it must be specified.
L195: erase “were”
Response : It has been done
L 199: remove the brackets. They are unnecessary.
Response : It has been done
L233-241: was the essential oil recovered from the hydrolate mixed to the essential oil obtained by hydrodistillation?
Response : The studied essential oils were obtained primarily by hydro-distillation using a Clevenger-type apparatus. At the end of extraction, we noticed that an important amount of essential oil remained in the hydrolate that’s why we have used a second solvent extraction using the dichloromethane.
L. 238: removal of the extraction solvent.
Response : It has been done
L. 281 remove the full stop after “Parafilm”
Response : It has been done
L.336: actually the differences were quantitative, and not “notable”, as the chemical compounds detected were the same, as explained few lines below.
Response : It has been done
L. 346: the authors recommend the use of C. ambroisoides EOs against weeds and pests, nevertheless no information about the toxicity against humans are reported. Moreover, the authors specified that the EOs were characterized by orange color and unpleasant odor, therefore in my opinion they could be not easily applied in the food industry. I invite the authors to better explain and motivate their conclusion or to add some considerations about it.
Response : Thank you very much for your valuable suggestion. The information required has been added at the end of the conclusions as recommended. Please see the revised version.
The manuscript has been revised, nevertheless the English language used is very simple. I kindly invite the authors to use a greater care in the revision of the manuscript, particularly as regards English language and punctuation, before the next resubmission.
Response : It has been done
Please see the attachment

This manuscript is a resubmission of an earlier submission. The following is a list of the peer review reports and author responses from that submission.
Round 1
Reviewer 1 Report
Comments and Suggestions for Authors
The manuscript entitled “Chemical Profiling, Insecticidal and Phytotoxic Effect of Essential Oils from Leaves and Inflorescence of Moroccan Chenopodium ambrosioides (L.)” is well designed. However, there are some queries that should be taken into account in the revised manuscript.
Abstract should be improved and include the results of HPLC MS analysis.
Introduction part should be revised and add more literature about the essential oil and monoterpenes present in Chenopodium ambrosioides (L.)
Line no 126 The phyto-toxic effect of CAEOs on alfalfa, flax and radish seeds was assessed,
Author used scientific name of the plants and should be consistent throughout the manuscript.
Author should provide GCMS chromatograms of essential oil of leaves and inflorescence.
Author used retention indices provide chromatograms of C9 to C24.
Why the total% composition is less than 100 in table -1?
97.54 and 95%
GCMS method should be more elaborate and properly.
Why author used Alfalfa 'Medicago sativa', Flax 'Linum usitatissimum' and Radish 'Raphanus sativus' seeds for the phytotoxic study
There are some typos error?
Why author not performed antimicrobial activity for the essential oil of leaves and inflorescence.
Add the new reference as mention below.
Essential oil from Chenopodium ambrosioides L.: secretory structures, antibacterial and antioxidant activities Acta Scientiarum. Biological Sciences, vol. 38, no. 2, pp. 139-147, 2016.
Author Response
Reviewer 1:
Thank you very much for taking the time to review this manuscript. Please find the detailed responses below and the corresponding revisions/corrections highlighted/in track changes in the re-submitted files.
The manuscript entitled “Chemical Profiling, Insecticidal and Phytotoxic Effect of Essential Oils from Leaves and Inflorescence of Moroccan Chenopodium ambrosioides (L.)” is well designed. However, there are some queries that should be taken into account in the revised manuscript.
- Abstract should be improved and include the results of HPLC MS analysis.
Response: The results of GC-MS analysis are included in the abstract section.
- Introduction part should be revised and add more literature about the essential oil and monoterpenes present in Chenopodium ambrosioides (L.)
Response : Thank you very much for your valuable suggestion. The information required has been added in the introduction as recommended.
- Line no 126 The phytotoxic effect of CAEOs on alfalfa, flax and radish seeds was assessed, Author used scientific name of the plants and should be consistent throughout the manuscript.
Response : It has been done
- Author should provide GCMS chromatograms of essential oil of leaves and inflorescence. Author used retention indices provide chromatograms of C9 to C24.
Thank you for your constructive comment. Data relative to GC-MS were extracted from chromatograms for both leaves and inflorescence. The study was carried out in 2020. Unfortunately, we have lost such chromatograms because of a viral attack.
- Why the total% composition is less than 100 in table -1? 97.54 and 95%.
Response: Thanks for your constructive comments. The results of chemical composition demonstrated that there are certain minor compounds not identified or in trace form.
- GCMS method should be more elaborate and properly.
Response : It has been done
- Why author used Alfalfa 'Medicago sativa', Flax 'Linum usitatissimum' and Radish 'Raphanus sativus' seeds for the phytotoxic study
Response: The species seeds used in this study are a model species widely used to determine the phytotoxic activity of essential oils.
- There are some typos error?
Response: Typo mistakes were corrected throughout the text. Please see the revised version.
- Why author not performed antimicrobial activity for the essential oil of leaves and inflorescence.
Response : I agree with you, but the antimicrobial activity of Chenopodium ambrisoides essential oils is well documented. However, your suggestion can be an objective to further study in the future to compare the antimicrobial activity of leaves and inflorescence of this species.
- Add the new reference as mention below.
Essential oil from Chenopodium ambrosioides L.: secretory structures, antibacterial and antioxidant activities Acta Scientiarum. Biological Sciences, vol. 38, no. 2, pp. 139-147, 2016.
Response : It has been done.

Reviewer 2 Report
Comments and Suggestions for Authors
The authors of the manuscript entitled “Chemical Profiling, Insecticidal and Phytotoxic Effect of Essential Oils from Leaves and Inflorescence of Moroccan Chenopodium ambrosioides (L.)” report in a clear and essential way the results of a study well described from an experimental point of view, however, a in my opinion, it should be better explained and underlined why the authors decided to test the two essential oils against the 3 selected plants and against Tribolium confusum, since the motivation does not emerge from the text.
Here are some other suggestions:
- chenopodium must have a capital C, throughout the manuscript
- Line 101-102: even if it is a repetition, the authors must specify that they are talking about CAEOs of leaves and inflorescences
- Line 127: given that the acronyms of the germination and growth parameters appear here for the first time, they should be explained at this point in the text
- Table2-4: the authors state "different letters indicate a significant difference (p˂0.05)", I think it is appropriate to clarify what a, b, and c refer to
- The sentence on lines 171-172 needs to be reviewed...apparently a verb is missing
- Line 223: “usedis” must be separated (check the whole manuscript, there are several words attached)
- the bibliographical notes are not very recent... it would be appropriate to replace at least some of them with works from the last maximum 5 years
Comments on the Quality of English LanguageThe English of the manuscript does not require editing, except for some grammatical errors and minor changes.
Author Response
Reviewer 2:
Thank you very much for taking the time to review this manuscript. Please find the detailed responses below and the corresponding revisions/corrections highlighted/in track changes in the re-submitted files.
The authors of the manuscript entitled “Chemical Profiling, Insecticidal and Phytotoxic Effect of Essential Oils from Leaves and Inflorescence of Moroccan Chenopodium ambrosioides (L.)” report in a clear and essential way the results of a study well described from an experimental point of view, however, a in my opinion, it should be better explained and underlined why the authors decided to test the two essential oils against the 3 selected plants and against Tribolium confusum, since the motivation does not emerge from the text.
Response : Thank you very much for carefully reviewing the details of our manuscript. At the end of the introduction section, we have cited as a principal objective of this study is to evaluate for the first time the variation in yield, chemical composition, insecticidal and phytotoxic activities of essential oils isolated from the leaves and inflorescences of Chenopodium ambrosioides.
Here are some other suggestions:
- chenopodium must have a capital C, throughout the manuscript
Response : It has been done
- Line 101-102: even if it is a repetition, the authors must specify that they are talking about CAEOs of leaves and inflorescences
Response: As per your comment, this mistake was corrected.
- Line 127: given that the acronyms of the germination and growth parameters appear here for the first time, they should be explained at this point in the text
Response: It has been done
- Table 2-4: the authors state "different letters indicate a significant difference (p˂0.05)", I think it is appropriate to clarify what a, b, and c refer to
Response: the letters used in Tables 2-4 on the same column indicate if there is a significant difference between the values registered of each parameter. These letters are generated by the statistical software used in this study. We have revise for more clarity. Please see Tables 2-4 in the revised version.
- - The sentence on lines 171-172 needs to be reviewed...apparently a verb is missing
Response : It has been done
- Line 223: “usedis” must be separated (check the whole manuscript, there are several words attached)
Response : It has been done
- the bibliographical notes are not very recent... it would be appropriate to replace at least some of them with works from the last maximum 5 years
Response : It has been done

Round 2
Reviewer 1 Report
Comments and Suggestions for Authors
Author revised manuscript significantly but query 4 : Author should provide GCMS chromatograms of essential oil of leaves and inflorescence. Author used retention indices provide chromatograms of C9 to C24. Thank you for your constructive comment. Data relative to GC-MS were extracted from chromatograms for both leaves and inflorescence. The study was carried out in 2020. Unfortunately, we have lost such chromatograms because of a viral attack.
GCMS chromatogram is important and based on that standard and samples data of GCMS, you will report your results; as author mentioned that they lost this data then no validity of this data. Therefore, it cannot be suitable for publication.
Author Response
Dear Reviewer,
The authors valued your constructive comment and we totally agree that chromatograms are important to support the data presented in the tables (chemical profile). However, as we clarified in the first round, the data presented in our study are extracted from the lost chromatograms. The raw data extracted from original chromatograms are enclosed in the attached file.
